# High-Power Diode Laser Surface Transformation Hardening of Ferrous Alloys

**DOI:** 10.3390/ma15051915

**Published:** 2022-03-04

**Authors:** Artur Czupryński, Damian Janicki, Jacek Górka, Andrzej Grabowski, Bernard Wyględacz, Krzysztof Matus, Wojciech Karski

**Affiliations:** 1Department of Welding, Silesian University of Technology, Konarskiego 18A, 44-100 Gliwice, Poland; damian.janicki@polsl.pl (D.J.); jacek.gorka@polsl.pl (J.G.); bernard.wygledacz@polsl.pl (B.W.); wojciech.karski@polsl.pl (W.K.); 2Institute of Physics—CSE, Silesian University of Technology, Krasińskiego 8, 40-019 Katowice, Poland; andrzej.grabowski@polsl.pl; 3Materials Research Laboratory, Silesian University of Technology, Konarskiego 18A, 44-100 Gliwice, Poland; krzysztof.matus@polsl.pl

**Keywords:** laser surface transformation hardening, high-power diode laser, ferrous alloys, thermography, cooling rate

## Abstract

A high-power direct diode laser (HPDDL) having a rectangular beam with a top-hat intensity distribution was used to produce surface-hardened layers on a ferrous alloy. The thermal conditions in the hardened zone were estimated by using numerical simulations and infrared (IR) thermography and then referred to the thickness and microstructure of the hardened layers. The microstructural characteristics of the hardened layers were investigated using optical, scanning electron and transmission electron microscopy together with X-ray diffraction. It was found that the major factor that controls the thickness of the hardened layer is laser power density, which determines the optimal range of the traverse speed, and in consequence the temperature distribution in the hardened zone. The increase in the cooling rate led to the suppression of the martensitic transformation and a decrease in the hardened layer hardness. The precipitation of the nanometric plate-like and spherical cementite was observed throughout the hardened layer.

## 1. Introduction

Laser surface treatment methods have been intensively employed for the past three decades for the enhancement of surface properties of different cast iron grades [1,2,3,4,5,6,7,8,9,10,11,12,13]. Much research has been devoted to improving the tribological properties of cast irons via a laser surface transformation hardening (LSTH) process [10,11,12,14,15,16,17,18]. In contrast to conventional surface hardening processes such as induction and flame hardening, LSTH ensures excellent surface finish and very low distortion of the workpiece. In many cases, this eliminates any need for post-treatment finishing [19]. The precise control of the heat input together with the ability to heat localized areas make LSTH a unique technology for the fabrication of locally selected wear-resistant hard surfaces on complex-shaped components [20]. In recent years, there has been significant interest in the application of LSTH to increase the tribological performance of ductile cast irons (DCI) in both the as-cast condition and after austempering heat treatment [14,15,16,17].

Many researchers have reported on the significant increase in the wear properties of DCI after the LSTH process [14,16]. The tribological properties of the laser hardened layers on DCIs depend upon the layer hardness, which, in turn, is affected by the phase composition of the layer [16]. On the other hand, the microstructural characteristics of the laser hardened layer and also the layer thickness are directly determined by the thermal conditions in the hardened zone [17]. For successful laser hardening, it is necessary to form a relatively homogeneous austenite structure in the hardened zone during a very short period of the laser-induced thermal cycle. This requires notably higher temperatures on the hardened surfaces than those during conventional hardening to achieve a sufficiently high diffusion rate for the homogeneous austenitic structure [18]. As a result, to obtain the desired layer thickness and to avoid partial melting, precise control of the surface temperature is needed. However, scant data are available on the effect of LSTH parameters on the thermal conditions during the processing of DCIs and the resulting microstructure of the hardened layers.

The main objective of this work was to assess the effect of the LSTH parameters on the thermal conditions in the hardened zone and the resulting change in the thickness and the microstructure of the surface-hardened layer. The LSTH process was performed on a pearlitic DCI using a high-power direct diode laser having a rectangular beam with a top-hat intensity distribution. The thermal conditions in the hardened zone were determined by using infrared (IR) thermography and numerical simulations. The surface-hardened layers were characterized in terms of their surface finish, microstructure and hardness.

## 2. Materials and Methods

The chemical composition of the DCI (EN-GJS-700-2) used in this study is listed in Table 1. The DCI composition was measured using both a LECO GDS 500A optical emission spectrometer (LECO Corporation, St. Joseph, MI, USA) and a LECO CS125 Carbon Analyzer (LECO Corporation, St. Joseph, MI, USA). The microstructure of the as-received DCI was composed of a pearlitic matrix (ferrite content <15%) and graphite nodules with an average diameter of about 30 µm (Figure 1a). The average thickness of the cementite lamellae was 160 nm (Figure 1b). The hardness of the DCI in the as-received condition was 290 ± 15 HV0.5. The DCI specimens in the shape of a rectangular plate (40 × 30 × 10 mm) were ground to an average roughness (*R*_a_) of approximately0.05 µm. The surface roughness measurements were conducted using a Surftest SJ-210 portable surface roughness tester (Mitutoyo Corporation, Kawasaki, Kanagawa, Japan). Prior to obtaining the reflection measurements and the laser hardening, the test plates were cleaned in acetone.

The reflectivity of the DCI specimens at room temperature was measured using a setup comprising an Ocean Optics PC2000-ISA-PC Plug in Fiber Optics Spectrometer (wavelength range 500–1000 nm; resolution: ±0.5 nm), an ISP-REF Integrating Sphere and a WS-1 diffuse reflectance standard (Ocean Optics Inc., Dunedin, FL, USA). Reflectance measurements were made using both diffuse and specularly reflected light.

The thermographic measurements were carried out with the use of a FLIR A655SC thermal camera (Teledyne FLIR LLC., Wilsonville, OR, USA). The camera works in the 7.5–14.0 µm wavelength range and a 16-bit dynamic range at a 50 Hz polling rate and has an accuracy of ±2% of reading.

The hardening trials were carried out with a high-power direct diode laser (HPDDL) with a rectangular beam with a top-hat intensity distribution in the slow-axis direction (HPDDL, Rofin-Sinar Laser GmbH, Hamburg, Germany). The dominant wavelength of HPDDL used was 808 nm. The HPDDL beam spot size was 1.5 × 6.6 mm (length × width). The focal plane of the laser optics was located at the DCI specimen and the slow HPDDL beam axis was set to be perpendicular to the scanning direction. Argon (at a flow rate of 10 L/min) was used during all hardening trials to prevent oxidation of the surface. Additional details of the HPDD laser source have been previously given [21]. The selected hardening conditions are listed in Table 2. The selection of the optimal range of processing conditions was aimed at avoiding the finish grinding operation in potential industrial applications. For this reason, the occurrence of partial surface melting was not accepted.

Single-pass hardened beads (SHBs) were examined by optical microscopy (Eclipse MA100, Nikon Instech Co., Ltd., Tokyo, Japan), scanning electron microscopy (SEM, ZEISS SUPRA 35, Carl Zeiss Microscopy GmbH, Jena, Germany), transmission electron microscopy (FEI Titan 80/300, Scientific and Technical Instruments, Hillsboro, OR, USA) and X-ray diffraction (XRD). SEM investigations were conducted using both secondary electrons and in-lens imaging modes. For both the SEM and optical investigations, the specimens were etched with 4% Nital. The thickness of the hardened layer was measured using Nikon image processing software on optical and SEM images taken from the SHB cross-sections. The TEM investigations were performed using the conventional TEM and scanning transmission electron microscopy (STEM) modes. STEM observations were carried out using a high-angle annular dark-field (HAADF) detector. TEM samples were prepared using an FIB technique using an SEM/FIB DualBeam microscope (FEI Helios NanoLab 600i FIB, Scientific and Technical Instruments, Hillsboro, OR, USA).

The phase composition of the hardened layers was examined by using a PANalytical X’Pert PRO MPD X-ray diffractometer (Malvern Panalytical, Almelo, The Netherlands). XRD patterns were obtained using Co-K*α* radiation (*λ* = 0.179 nm).

The hardness measurements were performed with a Wilson Wolpert 401 MVD microhardness intender (Wilson Wolpert Instruments, Aachen, Germany). The measurements on the hardened surface and the polished SHB cross-sections were conducted using 500 g and 100 g loads, respectively.

The laser hardening process was simulated by means of a nonlinear 3D transient thermal simulation in Visual Weld 16 (Sysweld core-ESI Group). The material database consisted of specific heat and thermal conductivity over the range of 20 °C to the melting point. The simulations take into consideration solid-state transformation with the Johnson–Mehl–Avrami–Kolmogorov equation for diffusive transformations and the Koistinen–Marburger equation for non-diffusive transformation. The equation parameters were chosen to fit the simulated TTT diagram to the experimental TTT diagram. A 3D Model with 691,620 elements and 187,640 nodes was developed (Figure 2). The dimensions of the developed 3D model were consistent with those of the specimens used in the experimental stage. The model was prepared in such a manner that where the simulated laser processing took place, hexa elements with a 0.05 × 0.05 × 0.05 mm size were used, and the rest of the model was composed of tetra elements with element size increasing with the distance from the processed area. Only half of the planar symmetrical sample was simulated to decrease computational complexity. This did not affect the simulation accuracy. The thermal boundary conditions were set to free air with a temperature of 20 °C exchange on the outer surfaces of the model excluding the symmetry plane. The process was simulated with the usage of a modified load 3D double ellipsoidal heat source proposed by Kik [22]. The double ellipsoidal heat source dimensions were calibrated to achieve the 800 °C isotherm dimensions equal to the dimensions of the hardening zone. The heat source parameters were as follows: length of 1.6 mm, depth of 0.1 mm and load zone width of 6.6 mm. The results of reflection measurements (Figure 3) were taken into account during the selection of energy input. The final heat source efficiency was in the 0.5–0.6 range.

## 3. Results

### 3.1. Surface Condition Analysis

Figure 4 and Figure 5 show the appearance of the laser hardened surface of the selected DCI in different processing conditions. The results of the hardness measurements on the hardened surfaces are listed in Table 2. The laser power level, determining the laser power density, has a direct impact on the optimal range of the traverse speed. With an increasing laser power density, the lower limit of the traverse speed, which ensures avoidance of partial melting, is notably increased (Table 2). The partial melting occurred in the regions directly adjacent to the graphite nodules and led to the formation of ledeburite eutectic structures on the processed surface (Figure 4a and Figure 5a). The presence of ledeburite eutectic regions increased the overall surface hardness of the processed layers (Table 2). Note that, to avoid finish grinding operations, surface melting is undesirable in transformation hardening. The surface hardness of the SHB fabricated in the optimal range of parameters (providing no surface melting) was in the range of 755 to 684 HV0.5. It should be noted that with increasing traverse speed, the maximum surface hardness decreased. To understand the effect of the processing conditions on the microstructure and the hardness of the laser surface-hardened layers on DCI, the next sections present detailed investigations on the microstructure and hardness of the SHBs produced under processing condition no. H2 and H8, as well as the thermal conditions in the hardened zone under those two processing conditions (Table 2).

### 3.2. Macro and Microstructural Analysis

Figure 6a,b show the cross-sectional macrographs of the SHBs produced under processing conditions no. H2 and H8, respectively (Table 2). The corresponding low magnification optical micrographs of the central area of the SHB cross-sections are presented in Figure 7. Optical microscope images taken from the undersurface area of the above-mentioned SHBs are shown in Figure 8. XRD patterns for the investigated DCI in the as-received condition and after the LSTH under processing conditions no. H2 and H8 are presented in Figure 9a–c, respectively.

The micrographs in Figure 7 suggest that the thickness of the hardened layer was approximately 60 and 200 µm for conditions no. H2 and H8, respectively. Based on the optical micrographs presented in Figure 8, one can observe that the microstructure of the hardened layer is composed of graphite nodules and the martensitic matrix. The XRD analysis indicated the additional presence of the retained austenite phase in the matrix of the hardened layer produced under processing condition no. H8 (Figure 9c). Note that the retained austenite fraction in the hardened layer produced under processing condition no. H2 was negligible (Figure 9b). Moreover, XRD patterns from all hardened layers gave peaks that can be identified as belonging to the cementite phase.

SEM investigations (Figure 10 and Figure 11) indicated that the hardened layer can be divided into two sub-regions: the region of complete dissolution of cementite lamellae and the region of partial dissolution of cementite lamellae. Directly beneath the hardened surface, there was a region exhibiting a complete dissolution of cementite lamellae (Figure 11a). The thickness of this region was approx. 22 and 135 µm in the SHB no. H2 and H8, respectively. The microstructure of this region was composed of the graphite nodules and the martensitic matrix (the martensitic/austenitic matrix in the case of SHB no. H8). The region of the hardened layer adjacent to the core material had a matrix that was, in general, composed of partially dissolved cementite lamellae and fine martensite laths (Figure 11b,c). The thickness of the region of partial dissolution of cementite lamellae was approx. 39 and 61 µm in the SHB no. H2 and H8, respectively.

TEM investigations of the region of complete dissolution of the cementite lamellae in the hardened layer (Figure 12 and Figure 13) revealed the presence of nanometric plate-like and spherical precipitates of cementite. The thickness of the cementite plates was close to 40 nm (Figure 12c). Both types of nanometric cementite precipitates were observed at the martensite lath boundaries. Further research will be required to clarify the mechanism of cementite precipitation during the investigated LSTH process.

### 3.3. Hardness Analysis

The hardness depth profiles of the SHBs no. H2 and H8 are reported in Figure 14. In general, the hardness profiles correspond well to the microstructural changes discussed in Section 3.2 and confirmed the previously mentioned thicknesses of the hardened layer. Moreover, the hardness test data confirmed the decrease in the surface layer hardness with increasing traverse speed mentioned in Section 3.1. Note that the subregions of the hardened layer, i.e., the regions of complete and partial dissolution of the cementite lamellae, have comparable hardness. This leads to a uniform hardness distribution throughout the hardened layer. The average hardness values for the SHBs no. H2 and H8 were 755 ± 5 and 651 ± 15 HV0.1, respectively.

### 3.4. Examination of Thermal Conditions in the Hardened Zone

The temperature distributions on the surface of the hardened specimens recorded by the IR thermography and calculated for processing conditions no. H2 and H8 (Table 2) are compared in Figure 15. The corresponding temperature profiles across the hardened zone are shown in Figure 16 and Figure 17, respectively. Figure 16 and Figure 17 show that the calculated temperature profiles match well with that recorded by the IR thermography, indicating the reliability of the numerical model used. Figure 18a,b present the calculated temperature fields in the cross-sections of the SHB no. H2 and H8, respectively. The calculated temperature distributions beneath the surface of the SHB under the above processing conditions are presented in Figure 19.

The thermal data indicated that the maximum temperature on the hardened surface (on the SHB centerline) was about 1000 and 1100 °C for processing conditions no. H2 and H8, respectively. The average heating rate for the above two cases was 2300 and 8500 °C/s, respectively. The time spent above the critical temperature (i.e., the matrix structure becomes entirely austenitic; assuming approximately 800 °C) for processing conditions no. H2 and H8 was 0.33 and 0.14 s, respectively. The average cooling rate within the temperature range of 800–500 °C for processing conditions no. H2 and H8 was 1300 and 3400 °C/s, respectively. The thermal data errors due to simplification of the modelled process, limited mesh density, idealisation of the simulation and variety of emissivity of the sample can be present. Although small deviations between the simulation and thermographic experimental measurements are present, their convergence is high, which increases the confidence of investigated thermal parameters.

## 4. Discussion

The laser power density, determining the optimal range of the traverse speed (ensuring no surface melting), had a direct impact on the hardened layer thickness. The power density for processing conditions no. H2 and H8 was 40.4 and 80.8 W/mm^2^, respectively. Despite a notably lower HI level (80 J/mm), the processing condition no. H8 provided an almost 3.5 times higher thickness of the hardened layers in comparison to that achieved under the processing condition no. H2 (HI = 120 J/mm). Similar relationships between the processing parameters and the hardening depth have been reported for LSTH of different grades of steel [23,24]. It should be noted that the maximum hardness produced in the hardened layer was affected by the traverse speed. The increase in the traverse speed led to a reduction in the maximum layer hardness. Comparing the surface hardness of the SHBs no. H2 and H8 shows that the lower traverse speed (0.2 and 0.6 m/min for H2 and H8, respectively) provided about 10% greater surface hardness (Table 2). This trend of decreasing hardness with increasing traverse speed is associated with an increase in the cooling rate and the resulting suppression of the martensitic transformation. The lower hardness of SHB no. H8 (651 HV0.1, Figure 14) in comparison to that of SHB no. H2 (755 HV0.1) is attributed to the presence of the retained austenite phase. The suppression of the martensitic transformation under non-equilibrium cooling conditions has been reported in several works on the casting and surface treatment of different cast iron grades [25,26]. It indicates that, from the point of view of the maximum surface hardness, the lower traverse speeds, ensuring lower values of the cooling rate, are beneficial in producing more wear-resistant layers [27,28]. On the other hand, to avoid surface melting with increasing power density, the elevation of traverse speed is required. The above results show that both the laser power density (determined by the laser power level and laser spot size used) and the traverse speed are equally important. Hence, a proper combination of the above processing parameters is required to achieve optimization of the hardened layer hardness and depth.

Considering the notably shorter thermal cycle induced under processing condition no. H8 (the time spent above the critical temperature for processing condition no. H2 was almost 2.5 times longer than that for H8), it can be concluded the main factor controlling the hardening depth under the analysed two cases is the value of the temperature on the hardened surface. This relationship is consistent with that previously reported for LSTH of steel [23,29,30]. The explanation can be found in the fact that the carbon diffusion coefficient is a strong function of the temperature. The structural changes in the hardened layer are dependent on diffusion-controlled processes such as the transformation of perlite to austenite and the homogenization of the carbon content in austenite. Based on the data reported by Gegner [31], it can be assumed that there is a significant increase in the carbon diffusivity in the face-centered cubic Fe-C austenite lattice with increasing temperatures in the range of 1000–1100 °C.

Comparing the microstructure of the SHBs (Section 3.2) with the calculated temperature distributions beneath the surface of SHBs (Figure 18 and Figure 19) suggests that the minimum temperature required for the complete dissolution of cementite lamellae in the hardened layer was about 970 °C. In turn, the formation of the hardened region that contained the partially dissolved cementite lamellae underwent treatment in the temperature range of 880–970 °C.

## 5. Conclusions

Surface-hardened layers were produced on ductile cast iron via laser surface transformation hardening. The laser source used was a high-power direct diode laser having a rectangular beam with a top-hat intensity distribution. The thermal conditions in the hardened zone were examined by using numerical simulations and infrared (IR) thermography. The results showed that the major factor controlling the thickness of the hardened layer is the laser power density, which determines the optimal range of the traverse speed and in consequence the maximum surface temperature. The hardening depth was found to increase with increasing power density. To avoid surface melting with increasing power density, the lower limit of traverse speed notably increases. The resulting increase in the cooling rate led to a suppression of the martensitic transformation and a decrease in the hardened layer hardness. In the investigated range of the laser power density (40.4–80.8 W/mm^2^), the maximum hardening depth was approx. 200 µm. The hardness in the hardened layer was increased by almost three times in comparison with the as-received condition.

## Figures and Tables

**Figure 1 materials-15-01915-f001:**
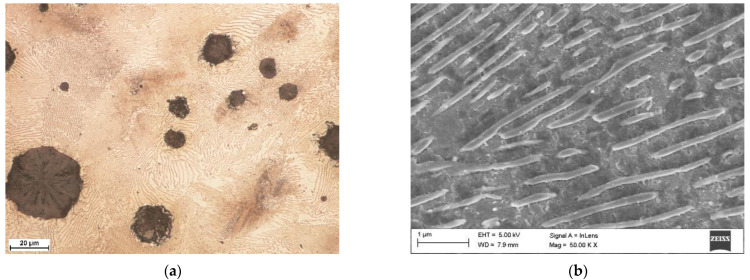
(**a**) Optical image showing the microstructure of the as-received DCI grade EN-GJS-700-2; (**b**) In-lens SEM image showing the perlite matrix.

**Figure 2 materials-15-01915-f002:**
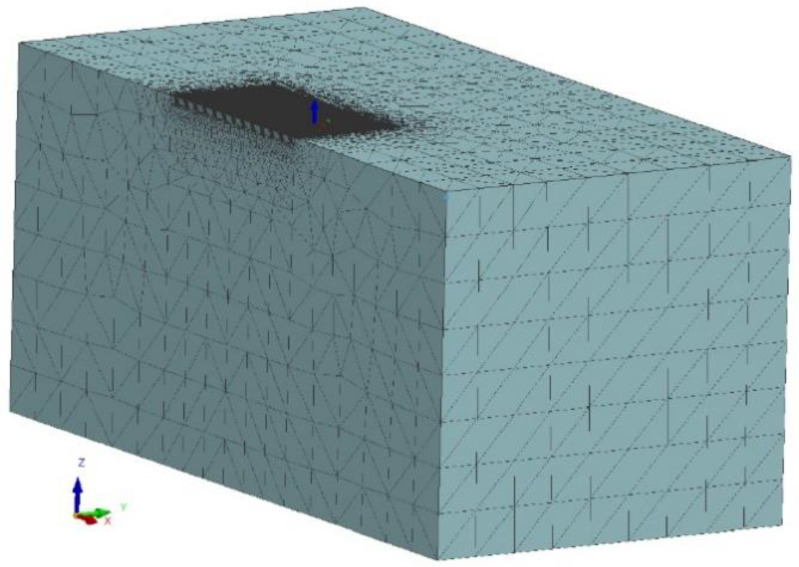
The developed 3D model and meshing arrangement.

**Figure 3 materials-15-01915-f003:**
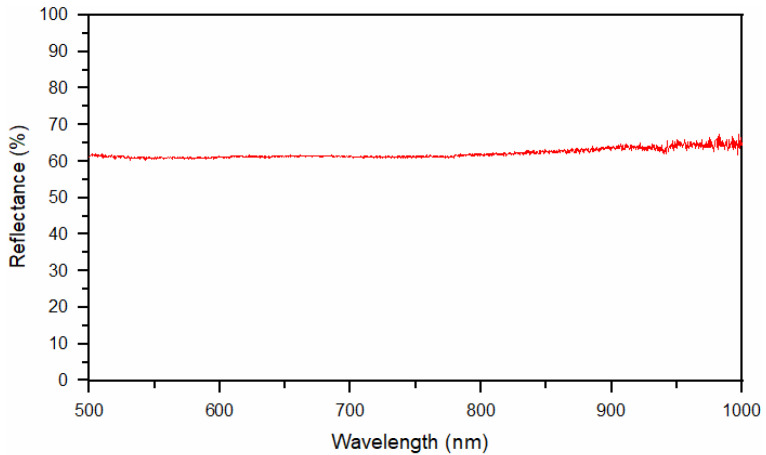
Spectral reflectance curve for a test plate of the DCI having an average roughness (*R*_a_) of approximately 0.05 µm.

**Figure 4 materials-15-01915-f004:**
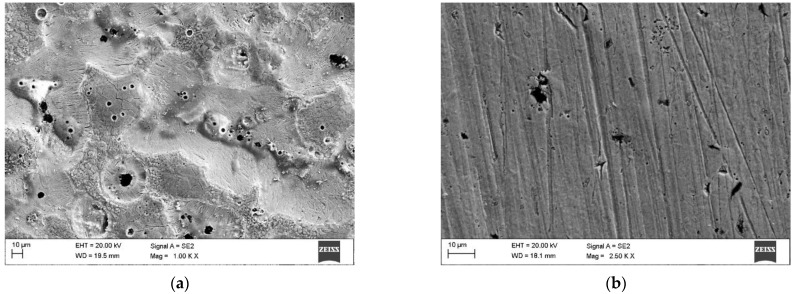
SEM images of the hardened surface at laser power of 400 W and traverse speed of: (**a**) 0.1 m/min; (**b**) 0.2 m/min.

**Figure 5 materials-15-01915-f005:**
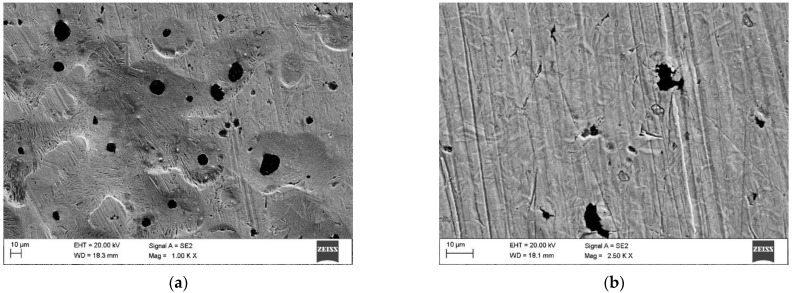
SEM images of the hardened surface at a laser power of 800 W and a traverse speed of: (**a**) 0.4 m/min; (**b**) 0.6 m/min.

**Figure 6 materials-15-01915-f006:**
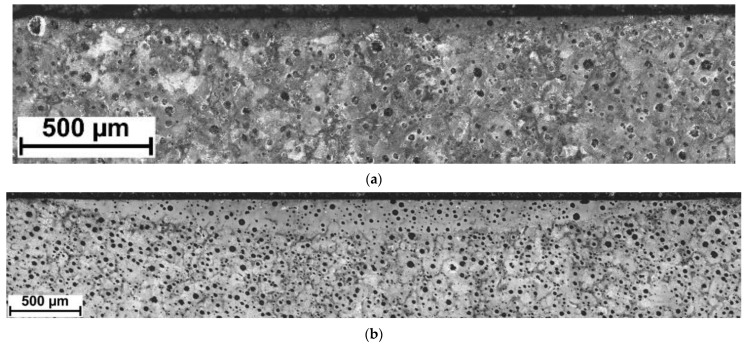
Optical macrographs of the SHBs no. (**a**) H2; (**b**) H8 (Table 2).

**Figure 7 materials-15-01915-f007:**
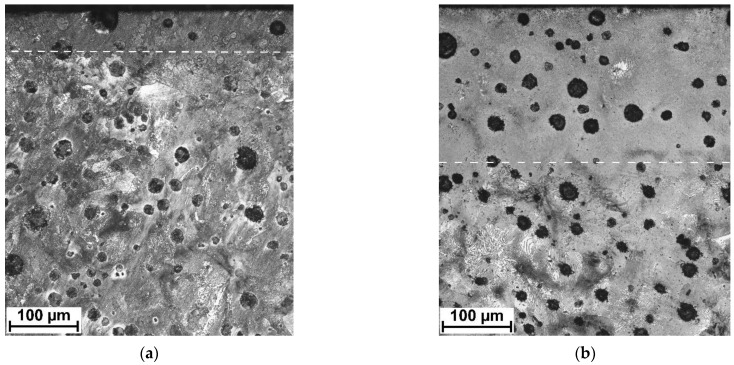
Low magnification optical micrographs of the SHB no. (**a**) H2; (**b**) H8 (Table 2). Dashed lines indicate an approximate boundary between the hardened layer and the core material.

**Figure 8 materials-15-01915-f008:**
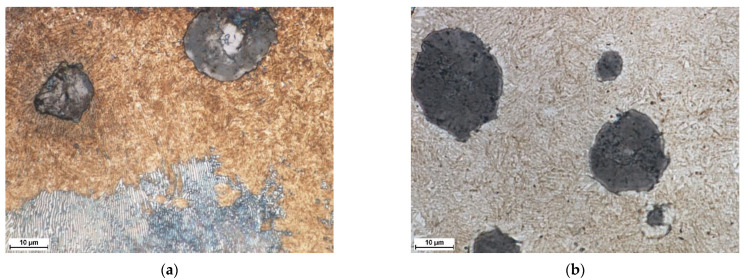
Optical micrographs showing the undersurface area of the SHB (in the center of the bead) no. (**a**) H2 (hardened layer and transition boundary) and (**b**) H8 (hardened zone) (Table 2).

**Figure 9 materials-15-01915-f009:**
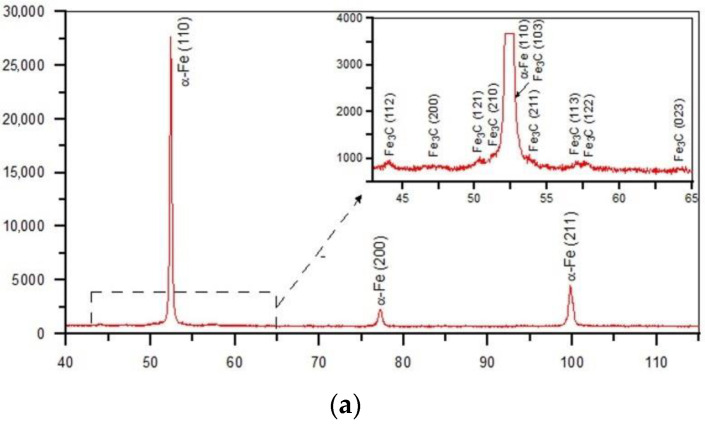
XRD patterns of the (**a**) as-received DCI and after the LSTH under conditions no. (**b**) H2 and (**c**) H8 (Table 2).

**Figure 10 materials-15-01915-f010:**
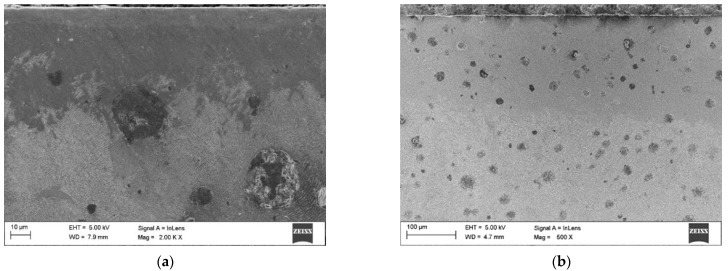
In-lens SEM images taken from the undersurface area of SHB no. (**a**) H2 and (**b**) H8 (Table 2).

**Figure 11 materials-15-01915-f011:**
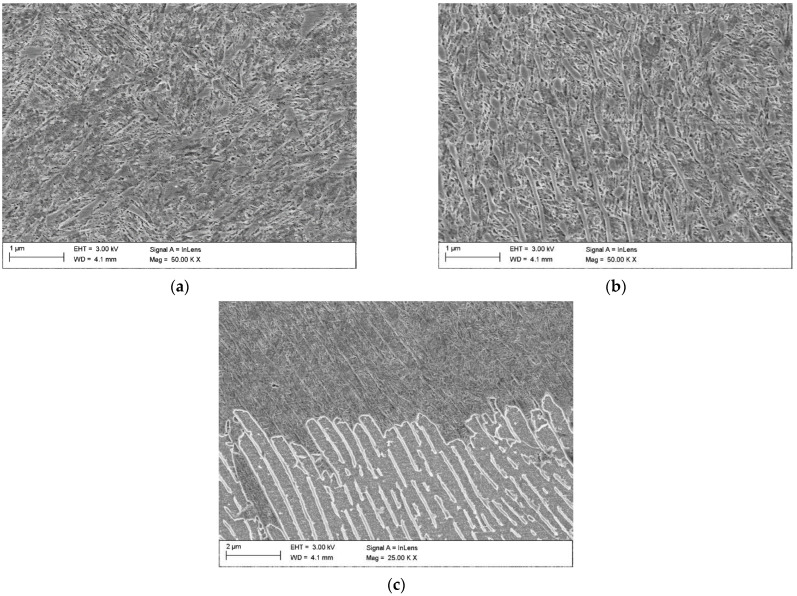
In-lens SEM images taken from the SHB no. H8 (Table 2) showing: (**a**) the region of complete dissolution of cementite lamellae; (**b**) the region of partial dissolution of the cementite lamellae; (**c**) a boundary between the region of partial dissolution of the cementite lamellae and the core material.

**Figure 12 materials-15-01915-f012:**
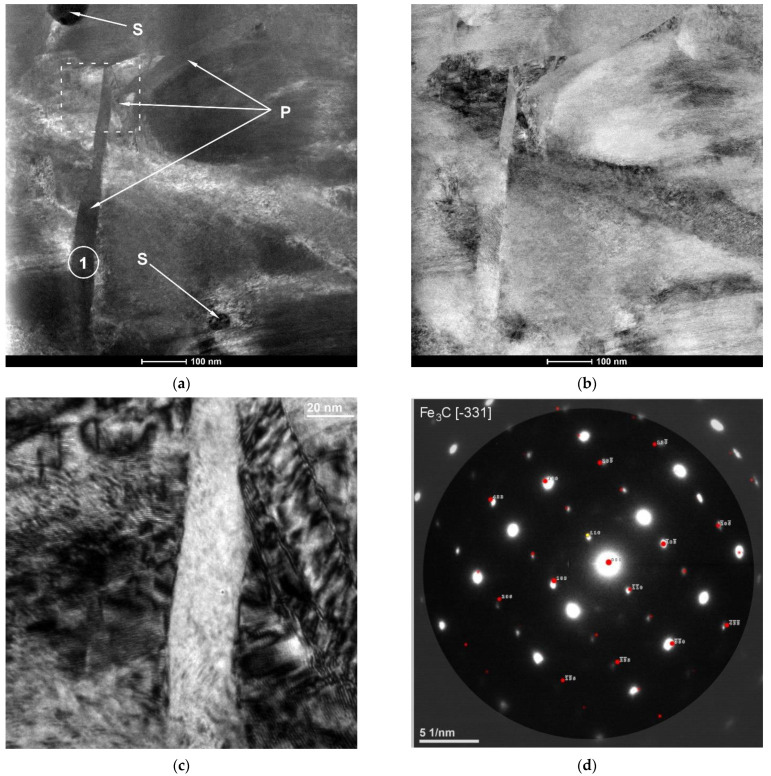
(**a**) Dark-field and (**b**) bright-field STEM HAADF images showing a distribution of plate-like (P) and spherical (S) cementite precipitates formed in the matrix of the SHB no. H8 (the region of complete dissolution of the cementite lamellae); (**c**) Bright-field TEM image taken from a marked area in (**a**); (**d**) SAED pattern obtained from region (1) in (**a**).

**Figure 13 materials-15-01915-f013:**
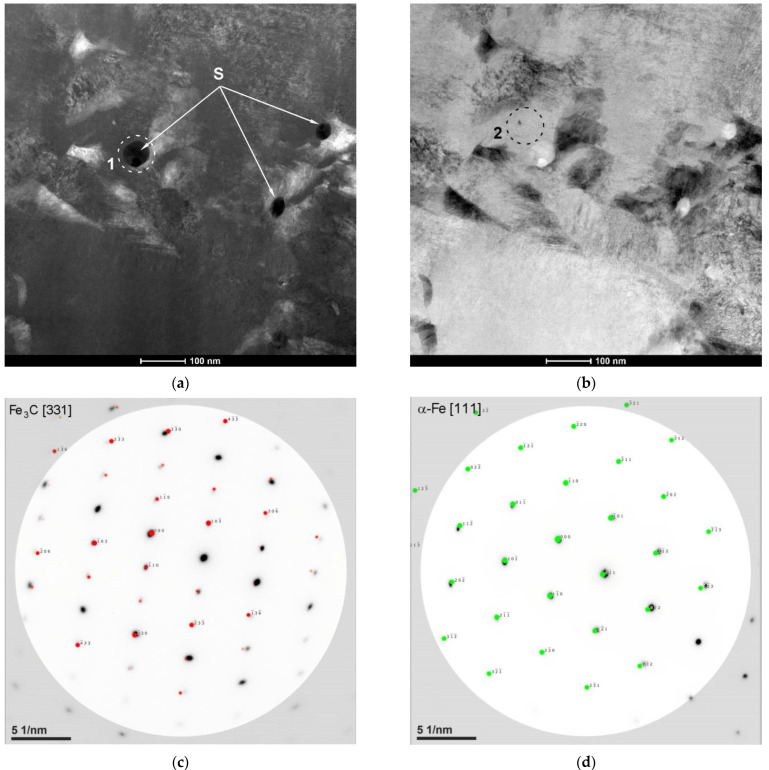
(**a**) Dark-field and (**b**) bright-field STEM HAADF images showing spherical (S) cementite precipitates formed in the matrix of the SHB no. H8 (the region of complete dissolution of the cementite lamellae); (**c**,**d**) SAED patterns obtained from regions (1) and (2), respectively, in (**a**).

**Figure 14 materials-15-01915-f014:**
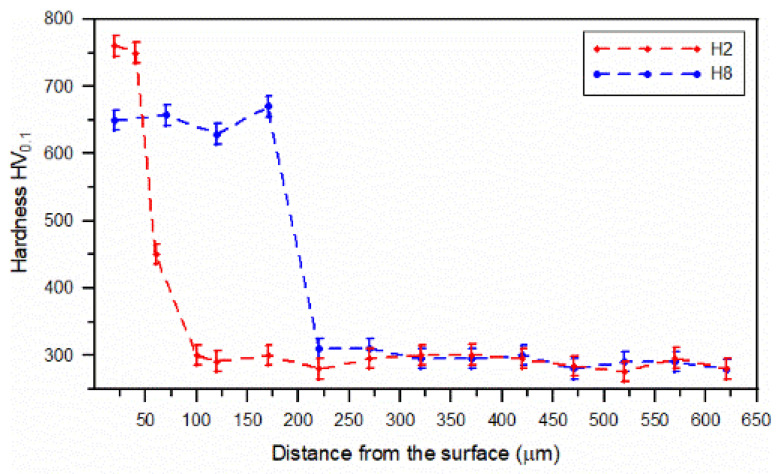
Hardness profiles of the SHBs no. H2 and H8 (Table 2).

**Figure 15 materials-15-01915-f015:**
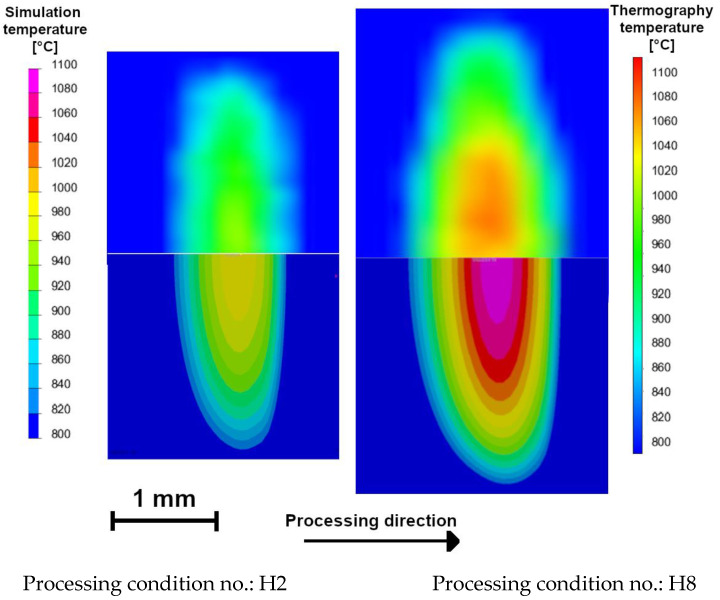
Temperature fields on the top surface of the laser surface-hardened specimens obtained via IR thermography (**upper part**) and thermal simulation (**lower part**). Processing conditions: H2 and H8 (Table 2). Selected emissivity value, *E* = 0.17.

**Figure 16 materials-15-01915-f016:**
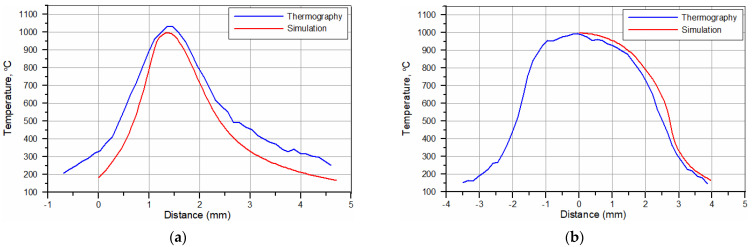
Temperature profiles for the hardened surface in (**a**) the longitudinal (along the SHB centerline) and (**b**) traverse directions (along the slow-axis of the HPDDL beam). Processing conditions no. H2 (Table 2).

**Figure 17 materials-15-01915-f017:**
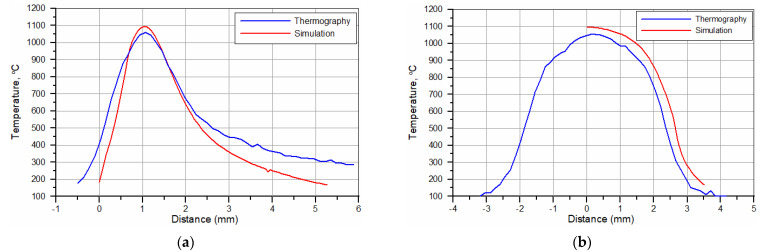
Temperature profiles for the hardened surface in (**a**) the longitudinal (along the SHB centerline) and (**b**) traverse directions (along the slow-axis of the HPDDL beam). Processing conditions no. H8 (Table 2).

**Figure 18 materials-15-01915-f018:**
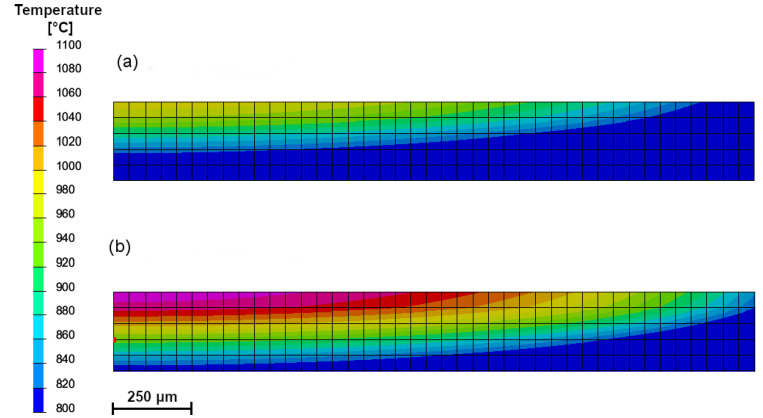
Calculated temperature fields in the cross-section of the SHB (a half of the bead) no.: (**a**) H2, (**b**) H8 (Table 2).

**Figure 19 materials-15-01915-f019:**
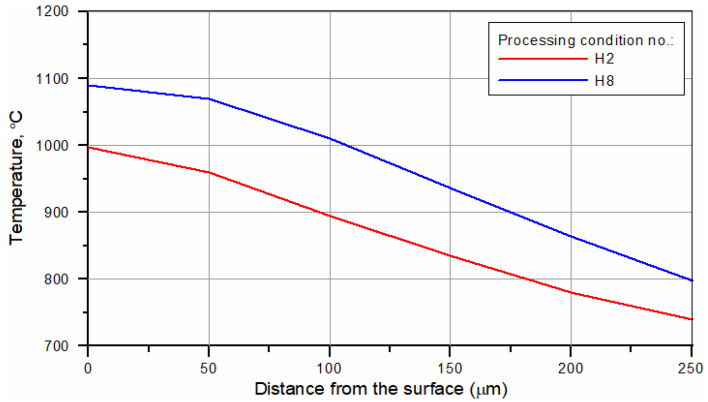
Calculated temperature distributions beneath the surface of the SHB (along the central vertical plane) under processing conditions no. H2 and H8 (Table 2).

**Table 1 materials-15-01915-t001:** Chemical composition of the investigated DCI substrate (wt%).

C	Si	Cu	Mn	Cr	Ni	S	P	Fe
3.52	2.62	0.80	0.24	0.02	0.04	0.008	0.016	balance

**Table 2 materials-15-01915-t002:** Selected processing conditions of laser surface transformation hardening.

Processing Condition No./SHB No.	Laser Power (W)	Laser Power Density (W/mm^2^)	Traverse Speed (m/min)	Heat Input ^1^ (J/mm)	Surface Hardness HV0.5	Layer Thickness ^2^ (µm)	Remarks/Quality ^3^
H1	400	40.4	0.1	240	867 ± 60	458 ± 14	PM
H2			0.2	120	755 ± 32	61 ± 11	NM
H3			0.3	80	290 ± 15	-	NH
H4	600	60.6	0.2	180	841 ± 54	437 ± 15	PM
H5			0.4	90	724 ± 29	192 ± 11	NM
H6			0.6	60	689 ± 31	78 ± 10	NM
H7	800	80.8	0.4	120	796 ± 58	393 ± 16	PM
H8			0.6	80	690 ± 28	196 ± 12	NM
H9			0.8	60	684 ± 30	92 ± 11	NM

^1^ defined by the ratio of the laser power and the traverse speed; ^2^ measured in the center of SHB; ^3^ assessed in terms of the surface condition: PM—partial melting, NM—no surface melting, NH—no hardening.

## Data Availability

Not applicable.

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
