# Peer review of "High-Power Diode Laser Surface Transformation Hardening of Ferrous Alloys"

_materials, 2022, doi:10.3390/ma15051915_

Round 1

Reviewer 1 Report

Needed changes

  1. The modelling description lacks data regarding material properties, the heat source. Some more information about simulation controls would be useful in addition to meshing. Enough information must be provided so that the calculations may be repeated. How are phase transformations considered? Does the model use different parameters to distinguish solid and liquid? What about solid-state phase transformations? Could they impact the shape of the thermal fields and the calculated cooling rate? Did you use the reflectivity as the efficiency of the laser absorption?
  2. The parameters used to define the heat source are empirical and need to be calibrated to each dataset. The calibrated heat source for the conditions studied are not presented in the work.
  3. Table 2 should also include the beam spot size to avoid confusion.
  4. The methodology should explain that only conditions that showed hardening and avoiding melting have been studied.
  5. Details of the thermal imaging is lacking. What was the response rate of the thermal camera? Is it fast enough to capture the motion of the laser without significant blur? How was it calibrated? What emissivity did you use? What wavelength range is the camera accurate for and what is the corresponding error?
  6. The thermal imaging data in Figure 16 and 17 should have error bars accounting for the uncertainty of these measurements.
  7. You mention critical temperatures on line 257. This needs to be introduced and explained.

Suggested changes

The findings in the report would be improved if they could help construct a process map, however the different beam sizes used may be why this has not been done with the experimental data. This could be potentially achieved with the process model.

On Figure 16 and 17 b), the full simulation should be shown on both sides so that it is clearer to the reader to evaluate the symmetry of the measurements. The model assumes the laser is perpendicular to the surface, however this is not true for all of the component and the laser shape can be skewed.

It would be good to compare the precipitated cementite lamellae size in the heat affected zone with the size in the parent material.

Author Response

Dear Reviewer 1

I send the responses to your comments in the attached file.

Yours sincerely

Artur Czupryński

Reviewer 2 Report

The authors studied the laser surface treatment of ductile cast irons using high powder direct diode lasers. While the manuscript is generally well executed, there are several issues that should be addressed before further consideration for publication.

  1. Suggest to provide more information, eg. schematic, on the surface treatment process. For example, several variables, such as laser power and traverse speed, are used in the experiment, however, their significance are not clearly defined.
  2.  What are the assumptions and boundary conditions used for the model?
  3. Any discussion on the sources of deviation between the experiment and simulations?
  4. For the microstructure, any discussion with the non-surface treated samples? Are the results expected? 

Author Response

Dear Reviewer 2

I send the responses to your comments in the attached file.

Yours sincerely

Artur Czupryński

Round 2

Reviewer 1 Report

I misunderstood and thought you had changed the beam spot size in addition to other parameters, and this was preventing you from plotting the results and creating a process map. If you had two axis, traverse speed, and laser power density, you could try and identify the regions where you transition from no hardening to hardening, and from hardening to melting. The process map might be sensitive to changes in the spot size used, the surface roughness, and component geometry.

As the beam size is constant, please remove it from Table 2. When you state the beam spot size on line 92, describe its width and length when providing the two dimensions.

There is a typo in table 2 and it should read “Laser Power Density”, rather than “Laser Dower Density”.

I appreciate the thermophysical parameters and phase transformation model used are part of the software database and are likely proprietary and cannot be published.

Could incipient melting of cementite occur? The timescales for diffusion based phase transformations to occur are quite short.

On lines 87,130, and 280 what do BW1, BW2 and BW3 refer to?

Author Response

Dear Reviewer 1,

Thank you for your thorough review of our manuscript. We have corrected the manuscript considering the comments, and We provided the required explanations. All alterations were highlighted in red within the revised version.

  1. I misunderstood and thought you had changed the beam spot size in addition to other parameters, and this was preventing you from plotting the results and creating a process map. If you had two axis, traverse speed, and laser power density, you could try and identify the regions where you transition from no hardening to hardening, and from hardening to melting. The process map might be sensitive to changes in the spot size used, the surface roughness, and component geometry.

The process map is a good idea. However, we investigated a quite narrow range of processing parameters (especially in the case of power density) and, we think that the resulting map would not be complete.  Additionally, as you pointed out, the surface roughness has a direct impact on thermal conditions during laser hardening. Currently, we are planning to conduct laser surface hardening of DCI using a wider range of processing parameters (including different laser beam geometry and surface conditions). The results of the planned research will be used to create the process maps and will be published in a future manuscript. 

  1. As the beam size is constant, please remove it from Table 2. When you state the beam spot size on line 92, describe its width and length when providing the two dimensions.

 The beam size has been removed from Table 2.

The sentence has been modified:  “The HPDDL beam spot size was 1.5 × 6.6 mm (length × width).”

  1. There is a typo in table 2 and it should read “Laser Power Density”, rather than “Laser Dower Density”.

The typo has been corrected.

  1. I appreciate the thermophysical parameters and phase transformation model used are part of the software database and are likely proprietary and cannot be published.

Yes, the material database is commercial and cannot be further reproduced.

  1. Could incipient melting of cementite occur? The timescales for diffusion based phase transformations to occur are quite short.

In the optimal range of processing parameters, the melting of cementite did not occur. The dissolution of cementite lamellae underwent by the solid-state transformation. The time above critical temperature is quite short, but the carbon diffusion rate is notably affected by the temperature. 

  1. On lines 87,130, and 280 what do BW1, BW2 and BW3 refer to?

They are markings for comments added by Bernard WyglÄ™dacz that point to which of the reviewer’s remarks changes made to manuscript respond to. These markings won’t be visible in final published paper.

In the attached manuscript, I send a revised manuscript of responses to your comments and suggestions. I also attach a certificate of linguistic correctness of the content of the article.

Your sincerely,

Artur Czupryński

Reviewer 2 Report

NIL

Author Response

Dear Reviewer 2,

Thank you very kindly for accepting the article. I also attach a certificate of linguistic correctness of the content of the article.

Your sincerely,

Artur Czupryński
